# A Custom Regional DNA Barcode Reference Library for Lichen-Forming Fungi of the Intermountain West, USA, Increases Successful Specimen Identification

**DOI:** 10.3390/jof9070741

**Published:** 2023-07-12

**Authors:** Michael Kerr, Steven D. Leavitt

**Affiliations:** 1Department of Biology, Brigham Young University, Provo, UT 84602, USA; 2M.L. Bean Life Science Museum and Department of Biology, Brigham Young University, Provo, UT 84602, USA

**Keywords:** Illumina, internal transcribed spacer (ITS), metabarcoding, species hypothesis, OTUs, taxonomic assignment, UNITE

## Abstract

DNA barcoding approaches provide powerful tools for characterizing fungal diversity. However, DNA barcoding is limited by poor representation of species-level diversity in fungal sequence databases. Can the development of custom, regionally focused DNA reference libraries improve species-level identification rates for lichen-forming fungi? To explore this question, we created a regional ITS database for lichen-forming fungi (LFF) in the Intermountain West of the United States. The custom database comprised over 4800 sequences and represented over 600 formally described and provisional species. Lichen communities were sampled at 11 sites throughout the Intermountain West, and LFF diversity was characterized using high-throughput ITS2 amplicon sequencing. We compared the species-level identification success rates from our bulk community samples using our regional ITS database and the widely used UNITE database. The custom regional database resulted in significantly higher species-level assignments (72.3%) of candidate species than the UNITE database (28.3–34.2%). Within each site, identification of candidate species ranged from 72.3–82.1% using the custom database; and 31.5–55.4% using the UNITE database. These results highlight that developing regional databases may accelerate a wide range of LFF research by improving our ability to characterize species-level diversity using DNA barcoding.

## 1. Introduction

DNA sequencing has revolutionized species delimitation and specimen identification [1,2,3]. Further innovations have emerged in the form of DNA barcoding, which uses variable genetic regions to identify individual specimens [4], and metabarcoding, which applies barcoding at multi-specimen scales [5,6]. Barcoding and metabarcoding success rely on DNA reference databases, to which sequences recovered during a study are compared [5]. The standard barcoding marker for fungi, the nuclear ribosomal internal transcribed spacer region (ITS) [7], is the most commonly sequenced genetic marker for fungi and is well-represented in public databases [8]. DNA barcoding of fungi using the ITS and presently available databases has transformed how fungal diversity is characterized [8,9,10,11,12,13].

Fungi are hyperdiverse, with recent estimates of total species ranging from 2.2 to 6.28 million [13,14]. DNA sequencing efforts over the past three decades have amassed millions of sequences representing a portion of the overall fungal diversity. The UNITE database includes curated, publicly available ITS sequences, providing more than 1,000,000 fungal ITS sequences for reference [8]. This database serves as a data provider for a range of barcoding pipelines and is used extensively in fungal-based barcoding and metabarcoding studies (e.g., [9,11,13]). However, less than half of the 140,000 described species are represented by DNA data [15], while the vast majority of estimated fungal diversity is not even represented by DNA data [8]. A significant percentage of UNITE’s species hypotheses may represent a part of this diversity, as 8.23% of species hypotheses created through clusters of ITS sequences at 1.5% dissimilarity, as of 2018, remain unidentified below the fungal kingdom [8]. This pattern of poor overall representation of biological diversity and low connection to established taxonomy is also seen in other lineages, including plants, arthropods, and chordates [16,17,18].

Limited representation of the total estimated fungal species diversity [14] in barcode reference databases leads to multiple practical issues. Inaccurate, unreliable, or otherwise ambiguous species-level identifications for specimens [19,20] hinder effectively connecting genetic diversity to known species. Sequences deposited in databases with unusable or erroneous taxonomic information also may encourage other scientists to deposit similarly unusable sequences [21]. Other consequences of unusable data and low species representation include hindered efforts in conservation concerns, since DNA barcoding can help in tasks such as tracking invasive species and estimating current biodiversity [22]. 

Metabarcoding relies on DNA collected from samples comprised of multiple taxa, either from environmental DNA or extracted material from said taxa [5], and often no vouchers are collected to link the specimen with the genetic data. Thus, metabarcoding studies rely on DNA reference databases that are built on sequences generated from vouchered specimens. Ideally, sequences represented in DNA reference libraries originate from expertly identified specimens [23], although in some cases, provisional names may be provided, and dynamic, updatable reference libraries can promote FAIR data practices [24].

Considering that species representation is lacking for lichen-forming fungi (LFF) and that metabarcoding needs a robust, voucher-backed database for reliability, we hypothesized that developing a regional-scale, voucher-backed DNA barcode library might help bridge the gap between current issues with DNA barcoding and future goals of reproducible and accurate sample identification. In the semi-arid Intermountain West of the United States, recent DNA-based research has revealed unexpected LFF diversity [25,26,27,28]. DNA metabarcoding has also been shown to recover significant genetic diversity within LFF species [29]. This diversity exists, in part, due to complex hydroclimatic variability, influenced both by topography and by atmosphere and ocean processes operating over a large range of temporal and spatial scales [30,31]. In this study, we construct a voucher-backed database of DNA barcodes from LFF species opportunistically collected throughout the Intermountain West. 

Using this custom, regional database, we assess its potential utility for improving specimen identification in LFF metabarcoding studies. We collected bulk community samples from 11 sites within the Intermountain West. From the bulk samples, we generated ITS2 metabarcoding amplicon data using Illumina short read sequencing. The metabarcoding samples were processed to assess species-level identification success, comparing the UNITE fungal database [8] with our custom regional database. Our study highlights the power of developing regional DNA reference libraries for LFF metabarcoding research.

## 2. Materials and Methods

Over the past 15 years, ITS sequence data from LFF have been amassed by the second author (SDL) from nearly 5000 specimens. These data originated both from published DNA-based research and opportunistic sequencing of the standard fungal DNA barcode not represented in published studies. Most sequences were generated from specimens collected throughout western North America, although a limited number came from specimens collected outside of the Intermountain West. From these sequences, we compiled a custom DNA reference database, hereafter referred to as the “LIMW database”. These sequences have recently been provided as a custom database in the Barcode of Life Data System (BOLD; https://boldsystems.org (accessed on 8 May 2023)) as BOLD project LIMW [29].

In the LIMW database, candidate LFF species—species hypotheses (SHs)—were delimited based on initial inferences from family-level molecular phylogenies with subsequent interpretations of vouchered specimens within a traditional taxonomic context [32]. Due to variable levels of taxonomic certainty in taxonomic determinations of SHs, see [33], a variety of naming conventions were applied. A qualitative phylogenetic species criterion was initially used to circumscribe species/SHs, and we did not implement empirical tree-based species delimitation approaches, such as [34]. Briefly, by taking into account phenotype-based determinations and a combination of branch support and branch length patterns, we required each SH to be monophyletic in family-level ITS phylogenies [35]. For sequences derived from specimens with high confidence determinations, we applied the coinciding taxonomic name. However, in cases where sequences representing a single species were not recovered as monophyletic or were inferred to comprise high levels of phylogenetic substructure in exploratory ITS topologies, these were treated as multiple SHs using “aff.” or “agg.” and the coinciding taxonomic name. In cases where sequences were derived from specimens with less certain determinations, we used “cf.” and the most appropriate coinciding taxonomic name. Finally, for sequences derived from unidentified specimens or ambiguous determinations, provisional species names were applied.

To assess the effectiveness of the LIMW database at improving species-level taxonomic assignments, we generated metagenomic data from bulk lichen community samples from eleven distinct sites in the Intermountain West (Table 1). For eight sites, bulk samples were collected from vouchered specimens collected as part of a regional lichen biomonitoring program in habitats in or near federally designated wilderness areas with diverse lichen communities. Small fragments of lichens were removed from the vouchered collections housed at the Herbarium of Non-Vascular Cryptogams (BRY-C) at Brigham Young University, Provo, UT, USA. In addition to the targeted lichen in the voucher, we also attempted to sample any additional co-occurring lichen thalli. Small fragments of lichens were removed using sterilized tweezers and placed directly into a Nasco Whirl-Pak sample bag. For the remaining three sites—Brigham Young University (BYU) campus, Delano Peak, and Paul Bunyan’s Woodpile—samples were collected in the field following [29]. The site at BYU represented an urban lichen community along a 100 m walkway with a conspicuous crustose lichen community. The Delano Peak site represented an alpine steppe site on a desert sky island, and the site at Paul Bunyan’s Woodpile comprised a basalt dike with a notably diverse saxicolous lichen community. Lichen communities at BYU and Paul Bunyan’s Woodpile were sampled by four student technicians and a professional lichenologist [SDL], and Delano Peak was sampled by SDL.

DNA was extracted from each of the 11 bulk community samples separately. Bulk samples (Table 1) were homogenized using sterilized mortar and pestles; and DNA was extracted from ca. two to four g. of homogenized material using the PowerMax Soil DNA Isolation Kit (Qiagen). From each bulk sample (meta-community DNA extraction), we amplified a portion of the ITS region [7]. Specifically, the hypervariable ITS2 region was amplified using polymerase chain reaction PCR with primers ITS3F (GCATCGATGAAGAACGCAGC) with ITS4R (TCCTCCGCTTATTGATATGC). PCR products were sequenced at RTL Genomics (Lubbock, TX, USA), using 2 × 300 paired-end sequencing on the Illumina MiSeq platform. 

FROGS v3.1 (Find, Rapidly OTUs with Galaxy Solution) was used to analyze ITS2 amplicon metabarcoding data. FROGS is a standardized pipeline containing a set of tools used to process amplicon reads produced from Illumina sequencing [10,37]. Paired-end reads for each sequence in the data were merged, primers were trimmed, and unmatched sequences were discarded in the preprocessing step. Merged reads were then filtered using the swarm clustering tool [38], and the clusters were first formed using the aggregation distance clustering set to 1. Rather than using a global similarity threshold, swarm uses adaptive sequence agglomeration. Therefore, clusters can be defined with extremely high precision, even with large datasets and is one of the most accurate clustering tools [39]. Chimeric sequences were then removed using the chimera removal tool implementing default parameters. Low abundance clusters were removed by setting the minimum proportion of sequences to keep operational taxonomic units (OTUs) to 0.000005 (from ca. 1000 total clusters), following [10]. All remaining clusters were filtered using the ITSx tool [40] to ensure that clusters met requirements for the ITS2 region in preparation for the taxonomic affiliation step. 

Initial taxonomic assignments of the OTUs were completed by comparing the sequences to the UNITE 8.0 database [8]. Only FROGS OTUs assigned to lineages comprised of lichen-forming fungi were considered further. For the UNITE assessment, LFF species-level assignment was considered successful when the BLAST percent identity of any OTU in that assignment was above either 98% or 98.5% to test the impact of two different sequence similarity thresholds. 

To assess the utility of the LIMW database, species-level identification of OTUs was considered successful if the OTU was recovered within a monophyletic clade with sequences representing a single SH from the custom BOLD database. Family-level multiple sequence alignments (MSAs) were generated from OTUs inferred from short read data from the 11 sampled sites and aligned with full length ITS sequences from BOLD project LIMW using the program MAFFT v7 [41]. We implemented the G-INS-i alignment algorithm and ‘1PAM/K = 2’ scoring matrix, with an offset value of 0.1, the ‘unalignlevel’ = 0.4, and the remaining parameters were set to default values. Family-level ITS MSAs were analyzed under a maximum likelihood (ML) criterion as implemented in IQ-TREE v2 [42], with 1000 ultra-fast bootstrap replicates [43], and the best-fitting substitution model for the entire ITS region selected using ModelFinder [44]. Trees were visualized using FigTree v1.4.4 [45]. No attempt was made to compile or include closely related ITS sequences from publicly available databases, e.g., NCBI’s GenBank. In family-level ITS phylogenies, OTUs recovered outside of species-level clades comprised of BOLD sequences were not considered as successful identifications. We note that using a phylogenetic species recognition criterion to infer taxonomic identity comes with significant caveats [32]. Finally, OTUs were collapsed into SHs based on the inferred family-level phylogenies.

Percentages of SH identification success, as well as associated means and standard errors for identification success and numbers of SHs identified, were calculated using a custom R script (Appendix A). These were calculated based on geographic sampling site and taxonomic family; total SH identification was also assessed for each reference library. Differences in the numbers of SHs identified in different treatments were analyzed for statistical significance using statistical tests run in R [46] with the custom R program. Successful SH identification differences among geographic sampling sites were shown to be normally distributed using the Shapiro–Wilk test. Successful SH identification differences among LFF families were shown via the Shapiro–Wilk test to not be normally distributed; however, a normal distribution was achieved after adding one to zero values and log-transforming the data. Normal distributions allowed for analysis through ANOVA. ANOVA tests were performed on both sets of data to determine any statistically significant differences in database performance. The Tukey HSD test was performed on both sets of ANOVA results as post hoc testing to determine which DNA reference libraries resulted in statistically significant differences in SH identification success.

Of the LFF families represented, nine had fewer than five SHs. These families were Graphidaceae, Peltigeraceae, Sporastatiaceae, Stereocaulaceae, Stictidaceae, Trapeliaceae, and three provisional family-level lineages of unknown taxonomic affinity. Of these families, the LIMW database had full, or 100%, representation for five, 50% representation for one, and no representation for three. UNITE had—at both the 98% and 98.5% thresholds—full representation for three, 50% representation for two, and no representation for four, and identified nothing in the other two. These nine families were excluded from statistical testing, so that outlier percentages would not skew results. These were included, however, in calculations of the total SHs identified to species and identification success by geographic sampling site.

## 3. Results

The LIMW database comprised 4862 ITS sequences and represented over 600 LFF species, including both formal taxonomic names and provisional species-level hypotheses. The database has been deposited in the Barcode of Life Data System (BOLD; https://boldsystems.org (accessed on 8 May 2023)) as BOLD project LIMW [29]. Short read data are available under the NCBI BioProject ID PRJNA972691. 

A total of 1007 OTUs were inferred from the ITS2 amplicon metagenomic short read data from the 11 sampled sites (Appendix A). Of these, 678 OTUs were assigned taxonomically to LFF lineages using the FROGS pipeline, and post hoc interpretation of the OTUs within a phylogenetic framework resulted in 473 LFF SHs after combining closely related and putatively conspecific OTUs (Appendix A). The number of SHs was variable between sites, with the lowest number, 65, found at Brigham Young University in Utah and the highest number, 146, found at Broom Canyon in Nevada (Table 1). 

Using the UNITE database, a total of 162 SHs were identified to species at a 98% sequence similarity threshold (34.2% of total SH count) and 134 at 98.5% (28.3% of total SH count). Three hundred and forty-two of the 473 SHs inferred here (72.3% of total SH count) were successfully assigned to species-level taxa represented in the custom DNA reference library, over two times the number identified by using UNITE (Figure 1).

Considering successful SH taxonomic assignments within the sampled families, the LIMW database outperformed the UNITE database (at both sequence similarity thresholds) except in the case of Cladoniaceae (Figure 2). Taxonomic family-based SH identification was highly variable, with no clear pattern among families; identification for the LIMW database, UNITE at 98%, and UNITE at 98.5% was between 25% and 88.9%, 0% and 83.3%, and 0% and 60%, respectively. Identification success rates using the LIMW database resulted in 23.6 successfully identified SHs on average, with ±5.51 SH standard error. UNITE identification success was generally lower than identifications using the LIMW database (35.9% ± 5.97% and 11.1 ± 2.65 SHs, 98% ID; 27.1% ± 4.27% and 9.07 ± 2.28 SHs, 98.5% ID). These numbers, however, were statistically insignificant (ANOVA, *p*-value 0.0636).

Identification of SHs with respect to site was less variable, ranging from 72.3% to 82.1% for the LIMW database; 34.7% to 55.4% for the UNITE database at a 98% sequence similarity threshold; and 31.5% to 47.7% for UNITE at 98.5% threshold (Figure 3). The mean percentage of identification and mean identification of species hypotheses (SHs) by site for the LIMW database was 77% ± 0.89% and 83 ± 6.10 species, respectively. Lower values of successful SH identification were observed using the UNITE database. At the 98% sequence similarity threshold, the mean percentage of successful identification per site was 42.5% ± 1.71%, while the mean species SHs identified per site was 44.5 ± 1.59. At the 98.5% threshold, the mean percentage of identification per site was 36.7% ± 1.43%, while the mean SHs identified per site was 38.5 ± 1.52. 

With respect to geographic sampling sites, the LIMW database outperformed UNITE for both thresholds (ANOVA, *p*-value 2.19 × 10^−9^). Performing a Tukey HSD post hoc test showed that the respective *p*-values for the LIMW database compared to UNITE at 98% and then to UNITE at 98.5% were, at most, 1 × 10^−7^. The Tukey HSD test for geographic sampling site showed that UNITE at 98% performed similarly to UNITE at 98.5% with a *p*-value of 0.491.

## 4. Discussion

In this study, we demonstrate the utility of developing custom, regional DNA barcode reference libraries to improve species-level taxonomic assignments for lichen-forming fungi (LFF) in metabarcoding research. Our voucher-based LIMW database (https://boldsystems.org (accessed on 8 May 2023); BOLD project LIMW) resulted in a more than a two-fold increase in the identification success of samples collected throughout the Intermountain West, USA, relative to the global UNITE fungal database [8] (Figure 2). Of the nearly 500 species/species hypotheses (SH) inferred from metagenomic data across the 11 sampled sites, 72.3% could be linked to species/SH in the LIMW database, while less than 35% of the total SHs were successfully assigned at the species level using the UNITE database. Our study demonstrates that developing regionally targeted ITS reference databases can help to significantly bridge the current gap in species representation in fungal DNA sequence databases [8]. Furthermore, the results suggest that the voucher-based LIMW database will likely have broad utility in LFF biodiversity research throughout the southwestern United States. Below we discuss the implications of our findings and highlight ways to improve this database in the future.

While other studies have created voucher-backed, regional-scale databases for LFF [47] and other organismal groups [48,49], to our knowledge, no studies have specifically addressed whether custom, regional-scale DNA reference databases improve species-level taxonomic assignments in metabarcoding studies. Focusing on restricted geographic and taxonomic scales, such as LFF in the Intermountain West in this study, likely increases the probability the researchers will reasonably be able to capture a broader range of relevant genetic and taxonomic diversity.

Even at limited geographic scales, lichen communities can be highly variable [50,51]. In the “Four Corners” region of the Intermountain West, USA, lichen diversity is only modestly predictable from climatic variables, and lichen communities are structured by nonlinear responses and interacting predictors [31]. These factors have resulted in the establishment of diverse lichen communities across the Intermountain West [25,26,28,52,53,54,55,56,57,58,59]. While characterizing differences among LFF communities in different habitats was not the aim of the present study, we documented limited overlap among different geographic regions sampled here (Figure 4). A significant proportion of LFF species-level diversity (roughly 25%) within each region in the Intermountain West was not shared with other regions (Figure 4). Interestingly, we found a core group of eight LFF species that are shared among the disparate regions—*Candelariella aurella*, *Lecidea stigmatea*, *Protoparmeliopsis muralis*, *Physciella chloantha*, *Rinodina* sp., *Rusavskia elegans* (two distinct SHs), and *Xanthomendoza montana*. The geographically diverse sampling sites across the Central Basin and Range ecoregion had the highest proportion of unique SHs (ca. 39%), followed by the only alpine habitat sampled here—Delano Peak (38%) (Figure 4). The only urban site sampled—Brigham Young University (BYU) campus, Provo, Utah—had a LFF community that was similar to “wild” sites in the Central Basin and Range ecoregion, the same ecoregion where BYU is located.

Notably, using the custom regional database resulted in relatively consistent success in the proportion of OTUs assigned to SHs across the 11 sampled sites using the LIMW database, ranging from 72.3% to 82.1% (Figure 3). These results suggest that the current reference library includes the majority of LFF lineages across a broad geographic area and diverse habitats in the Intermountain West. In contrast, we found wider variation in identification success when considering LFF families. Within LFF families, successful species-level assignment ranged from 0% to 100% identification (Figure 2). Families represented by fewer SHs showed more stochasticity in identification success rates, with consistently poor success using the UNITE database for families that were represented by fewer than five SHs inferred from our metagenomic data. Successful species-level assignments were highest in the families *Lecanoraceae*, *Physciaceae*, *Teloschistaceae*, *Umbilicariaceae*, *Psoraceae*, and *Verrucariaceae* (Figure 2), suggesting that diversity of these families is well-represented in the LIMW database. In contrast, members of *Ramalinaceae*, *Caliciaceae*, *Cladoniaceae*, *Candelariaceae*, and *Rhizocarpaceae* had the lowest rates of successful species-level assignments using the LIMW database. Future targeted taxonomic sampling of these families for vouchered specimens and associated ITS sequence data will be critical for efficiently increasing the taxonomic coverage in the custom regional database.

In this study, 25 of the 678 OTUs documented in the bulk community samples had low sequence similarity (<90% similarity) to sequence data presently available on GenBank. Many of these SH belonged to *Acarosporaceae*, crustose members of *Physciaceae*, *Megasporaceae*, and *Verrucariaceae* (Appendix A), lineages where taxonomic limitations are recognized. Most of the SHs from community bulk samples could not be linked to species/species hypotheses in the LIMW database. However, in cases where SHs could be linked to clades represented by vouchered specimens, those lichens have previously been recognized as problematic, e.g., species-level lineages in *Candelariales* and *Megasporaceae* species [25]. We also note that the cyanolichen family *Collemataceae* is very poorly represented in our metagenomic data and the LIMW database. This is likely due to challenges with amplifying the highly variable ITS region using traditional primers for members of *Collemataceae* (Bruce McCune, personal communication). Cyanolichens are sensitive to air pollution and other disturbances [60] and future work should be dedicated to developing methods to better capture this diversity using LFF metabarcoding approaches.

In addition to taxonomic and geographic sampling biases, there are a number of limitations inherent to metabarcoding approaches for fungi. First, in some fungi, intraspecific variation in the nuclear ribosomal cistron has been observed and/or potential intragenomic variation occurs among copies of the nuclear ribosomal cistron within a single genome [15,61]. This variation may lead to an increase in the number of sequence clusters using amplicon data, potentially resulting in artificially inflated species numbers [62]. Despite the possibilities of inflating species-level diversity, characterizing intraspecific genetic diversity remains important [29]. Second, DNA barcodes may not be sufficient for definite specimen identification [15]. In fungi, this limitation should consider ITS variability integrated with careful phenotypic evaluation (e.g., [63]). In practice, this integrative taxonomic approach treats each SH as a testable hypothesis, using empirical species delimitation methods to find the best supported species models [64,65,66,67]. Third, while all specimens in the LIMW database are backed by vouchered specimens, taxonomic determinations remain ambiguous in many cases. We anticipate that some of the ambiguously determined SHs will subsequently be associated with formally described species, while others represent diversity within species complexes or intraspecific variation. The amount of undescribed species-level diversity among the SHs in the LIMW database remains uncertain. Detailed taxonomic work is lacking for some of the vouchered specimens included in the LIMW database. Ongoing collaborations will be essential for updating the determinations and improving the taxonomy in our custom database.

Ultimately, for many LFF, phenotype-based species delimitation and specimen identification remains difficult [32,68,69,70]. In accordance with the FAIR guiding principles for optimizing scientific data and its reuse [24], findable and accessible vouchered specimens housed at BRY-C and associated sequence data available in BOLD facilitate subsequent inclusion of these data in future taxonomic revisions. The custom database is dynamic and can be easily updated to reflect the most up-to-date taxonomy, in addition to different analytical approaches for sequence-based species delimitation. Similarly, the short-read amplicon data generated for this study are publicly available, and we encourage researchers to continue to explore these data to identify best practices and novel perspectives into the LFF diversity in the Intermountain West.

In conclusion, our results show that our custom regional LFF DNA barcoding database can identify more OTUs from metabarcoding samples than a general database with broader scope. We suggest that regional databases perform well due to their focused nature—considering only restricted geographic regions and taxonomic groups. Developing voucher-based regional databases should be a top priority to facilitate future fungal biodiversity research. Using vouchered specimens to generate DNA barcodes is recommended [23] and the value of vouchered-based DNA reference libraries is well demonstrated [47,71,72]. We suggest that gaps in species representation in general databases can be filled more quickly and more thoroughly by focusing on these regional databases. At the same time, regional databases complement and can eventually supplement general databases as regional efforts are linked to global work, in the same way that we linked the LIMW database to BOLD.

## Figures and Tables

**Figure 1 jof-09-00741-f001:**
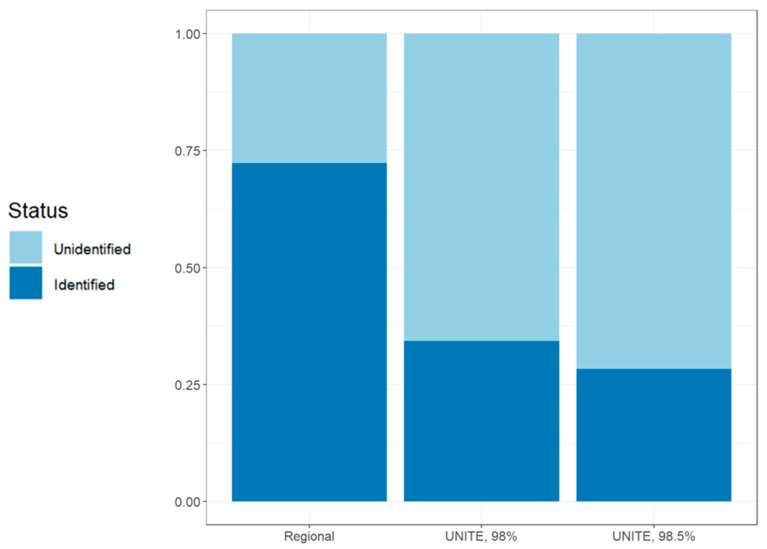
Species/species hypotheses (SH) identification success rates using the custom regional database (“LIMW database” on BOLD; https://boldsystems.org (accessed on 8 May 2023)) and the UNITE databases at two different sequence similarity thresholds. Each bar shows the proportion of identified versus unidentified species hypotheses (SHs) out of the total SHs. For all figures, “Regional” refers to the database made for this study; “UNITE __%” refers to UNITE as assessed through the BLAST Percentage ID statistic through the NCBI BLAST interface. The *x*-axis shows the database in question, while the *y*-axis shows the proportion of SHs identified to species-level.

**Figure 2 jof-09-00741-f002:**
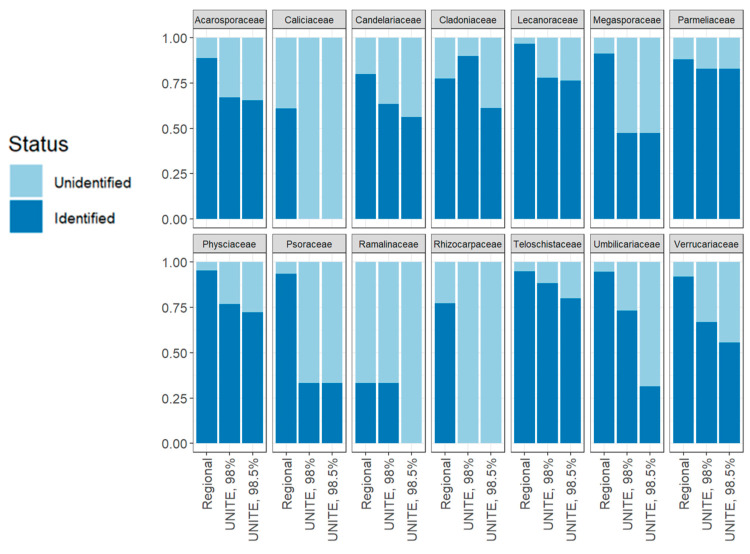
Species/species hypotheses identification success rate by family using the custom regional database (“LIMW database” on BOLD; https://boldsystems.org (accessed on 8 May 2023)) and the UNITE database at two sequence similarity thresholds. Each bar shows the proportion of identified versus unidentified species hypotheses when grouped in taxonomic families. The *x*-axis shows the database in question, while the *y*-axis shows the proportion (out of 1.00). Each facet graph is one taxonomic family.

**Figure 3 jof-09-00741-f003:**
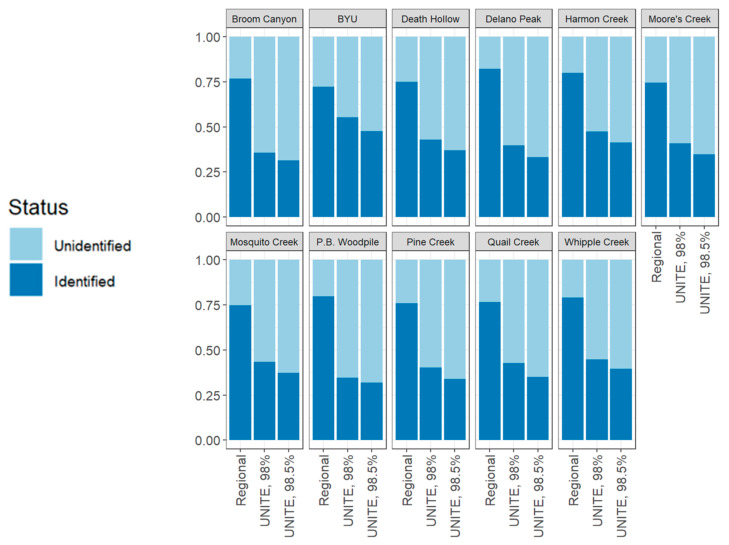
Species/species hypotheses identification success rate by site using the custom regional database (“LIMW database” on BOLD; https://boldsystems.org (accessed on 8 May 2023)) and the UNITE database at two sequence similarity thresholds. Each bar shows the proportion of identified versus unidentified species hypotheses when grouped by geographic sampling site. The *x*-axis shows the database in question, while the *y*-axis shows the proportion (out of 1.00). Each facet graph is one geographic sampling site.

**Figure 4 jof-09-00741-f004:**
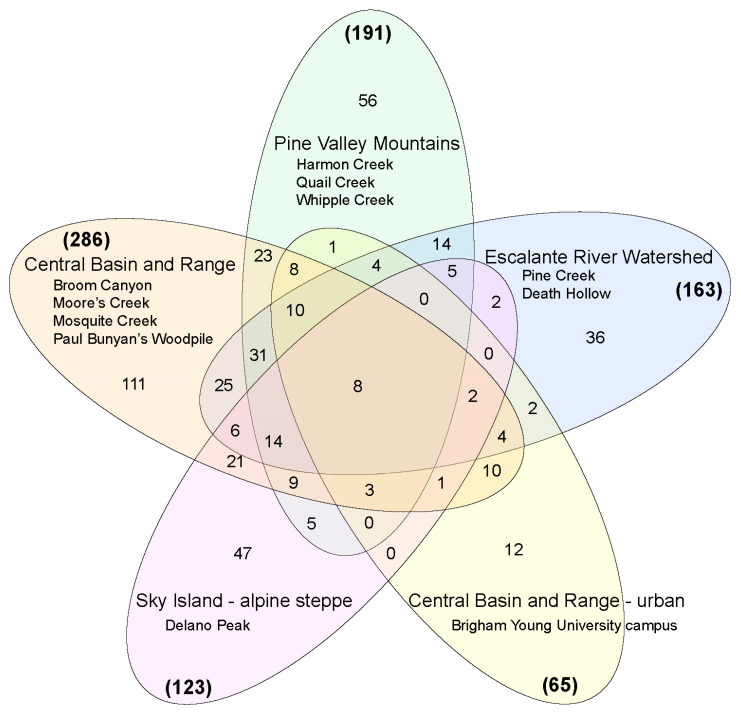
Venn diagrams comparing the overlap of species/species hypotheses among five regions: the “Central Basin and Range” ecoregion; an urban site (Provo, UT, USA) in the Central Basin and Range; the Pine Valley Mountains, a transition zone in southwestern Utah between the Colorado Plateau and Great Basin; the Escalante River Watershed in southern Utah (a major drainage into the Colorado River); and an alpine steppe habitat on a desert sky island in southern Utah. The total number of species/species hypotheses for each region is given in parentheses in bold text, and the number of shared species/species hypotheses is given for each comparison.

**Table 1 jof-09-00741-t001:** **Description of sites in the Intermountain West, USA, from which lichens were sampled for DNA metabarcoding.** Samples were used to compare the custom regional database to the UNITE database. Ecoregions are based on the classifications by the United States Environmental Protection Agency [36].

Site Name	Level 3 and 4 Ecoregions; Site Description; Latitude, Longitude, Altitude; Collection Date.	Sample Type	Species/OTUs
**Brigham Young University (Utah)**	Central Basin and Range/Moist Wasatch Front Footslopes Utah County, Brigham Young University Provo Campus, on cement wall and stairway, west of the McKay Building; 40.247, −111.652, 1400 m above sea level (m.a.s.l.); 5 June 2019.	Bulk	65/94
**Broom Canyon (Nevada)**	Central Basin and Range/Carbonate Woodland Zone Nye County, Humboldt-Toiyabe National Forest, White Pine Range (Currant Mountain Wilderness), Ely Ranger District, east of Railroad Valley, at mouth of Broom Canyon, west-facing slope of White Pine Peak, west side of Currant Mountain Wilderness Area; 38.890, −115.500, 2063 m.a.s.l.; 29 June 2011.	Bulk	146/171
**Moore’s Creek (Nevada)**	Central Basin and Range/Central Nevada Mid-Slope Woodland and Brushland Nye County, Humboldt-Toiyabe National Forest, Alta Toquima Wilderness, Moore’s Creek Trailhead; 38.860, −116.934, 2330 m.a.s.l.; 7 August 2013.	Vouchers	98/125
**Mosquito Creek (Nevada)**	Central Basin and Range/Central Nevada Mid-Slope Woodland and Brushland Nye County, Humboldt-Toiyabe National Forest, Table Mountain (Table Mountain Wilderness & vicinity), near boundary of Table Mountain Wilderness Area, along USFS Road No. 4409b, at Mosquito Creek Trailhead; 38.807, −116.682, 2210 m.a.s.l.; 8 August 2013.	Vouchers	115/145
**Paul Bunyan’s Woodpile (Utah)**	Central Basin and Range/Woodland- and Shrub-Covered Low Mountains Juab County, basalt dike ‘Paul Bunyan’s Woodpile’, near Jericho Junction; 39.767, −112.115, 2045 m.a.s.l.; 29 April 2019.	Bulk	144/188
**Pine Creek (Utah)**	Colorado Plateaus/Escarpments Garfield Co., Dixie National Forest, Box Death Hollow Wilderness Area, ~15 km north of Escalante along Hell’s Backbone Road (USFS Road No. 153), ~1.5 km north of “Box Trailhead” along USFS Trail No. 4009, along Pine Creek (collections made in riparian habitat along Pine Creek and surrounding sandstone outcrops); 37.865, −111.634, 1970 m.a.s.l.; 13 July 2015.	Vouchers	112/135
**Death Hollow (Utah)**	Wasatch and Uinta Mountains/High Plateaus/Wasatch and Uinta Mountains Garfield County, Dixie National Forest, Box Death Hollow Wilderness Area, at sandstone ridge south of head of Death Hollow, ~0.5 km southwest of Box Death Hollow Bridge, along Hell’s Backbone Road (USFS Road No. 153); 37.966, −111.599, 2661 m.a.s.l.; 13 July 2015.	Vouchers	100/121
**Delano Peak (Utah)**	Wasatch and Uinta Mountains/Alpine Zone Beaver/Piute Counties, Fish Lake National Forest, above tree line in alpine steppe habitat, vicinity of Delano Peak; 38.370, −112.376, 3650 m.a.s.l.; 16 September 2017.	Vouchers	123/159
**Harmon Creek (Utah)**	Wasatch and Uinta Mountains/Semiarid Foothills Washington County, Dixie National Forest, Pine Valley Mountains Wilderness Area, west of USFS Road No. 037, along USFS Trail No. 3028, vicinity of Harmon Creek Trailhead (riparian habitat along Harmon Creek); 37.364, −113.3518, 1764 m.ASL; 9 July 2015.	Vouchers	80/103
**Quail Creek (Utah)**	Wasatch and Uinta Mountains/Semiarid Foothills Washington County, Dixie National Forest, Cottonwood Forest Wilderness Area, Water Canyon, along Quail Creek, east of USFS Road No. 031, (Oak Grove Road), at spur road—USFS Road No. 4059, immediately south of private land (Sagewood Ranches); collections were made along Quail Creek and upslope in a Pinyon-Juniper woodland; 37.255, −113.427, 1212 m.a.s.l.; 11 July 2015.	Vouchers	103/122
**Whipple Creek (Utah)**	Wasatch and Uinta Mountains/High Plateaus Washington County, Dixie National Forest, Pine Valley Mountains Wilderness Area, east of Pine Valley, along Whipple Trail (USFS Trail No. 3025), east of wilderness boundary (riparian habitat along Middle Fork of Santa Clara River and upland Gambel oak-Mountain mahogany habitat); 37.368, −113.452, 2230 m.a.s.l.; 10 July 2015.	Vouchers	96/123

## Data Availability

The LIMW database has been deposited in the Barcode of Life Data System (BOLD; https://boldsystems.org (accessed on 8 May 2023)) as BOLD project LIMW [35]. Short read data are available under the NCBI BioProject ID PRJNA972691.

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
