# Peer review of "A Custom Regional DNA Barcode Reference Library for Lichen-Forming Fungi of the Intermountain West, USA, Increases Successful Specimen Identification"

_jof, 2023, doi:10.3390/jof9070741_

Round 1

Reviewer 1 Report

This is a nicely written, mostly clear paper, with a well evidenced thesis. My main concern is that the paper would be more readable if there authors were more careful with their descriptions of their methods, in particular around the use of the terms OTU and SH. Most readers will have only encountered SHs in the context of UNITE, though of course it is perfectly justified and reasonable to use the concept here and elsewhere. Could you add a tiny little flowchart figure outlining the steps or a sentence in the methods that really sharpens this up?

Please don’t use ‘LFF database’ throughout the paper, but rather, use its name ‘LIMW database’. After a while, I started to think of LFF as the name of your BOLD project.

49-51 Unclear. Please restate in simple terms. If more than half are not represented, how can the majority be even more poorly so?

51 This lack of identification of sequences merits a sentence, not a parenthetical. 8% is an important fraction, and the problem is worse in some groups than others, preventing the sequences being used as references. Outdated taxonomy is another big issue in these databases that is not mentioned. This unexpectedly comes up at L60. I think a little reorganization of this section would be helpful, with better paragraph structure. L63 reads as a topic sentence but is in the middle of a paragraph with a similar topic sentence – combine and re-jig.

76 define LFF here at first use in the text.

107 I think you need a little topic sentence here including the idea that the sequences were analysed in order to define species hypotheses, and, separately that these were delineated using a range of analyses and naming conventions.

109-110 Two e.g.’s suggest a re-write is needed here. Just say what you did and why rather than picking out a reference that you didn’t use

111 I think if you start this sentence with ‘Briefly,’, it becomes clear that this is the method you used to come up with your SH in the LIMW.

119 arbitrary species names? This sounds potentially fun – you mean nicknames? The way it’s written leaves us hanging and curious. Maybe give an example and choose a better word.

165 what is the number – or range of numbers of reads that that proportion represents across samples? The number that is here in line 165 doesn’t have units. Is it a proportion of total reads? Please rewrite this more clearly. Here it would be nice to briefly state the standard clustering of sequences for OTU calling in FROGS.

166 Please add the citation for ITSx

168 I did wonder what the consequences of using an initial UNITE filter would do for your OTU-to-LFF first pass step, given that its ability to determine your taxa was often so poor. Did some other LFFs slip through the cracks and is there a way to assess that?

191-2 At what levels – geographic, taxonomic, dataset – were the data aggregated for these tests?

195-6 are the data you are referring to here the numbers of taxa per site/area or family? Not clear which data are normally distributed. If you are concerned about meeting assumptions of a test, it might be worth a short explanation about that in terms of risks of failing these assumptions.

197 Repetitive: “For OTU identification comparing OTU identification success…”

206 What is a provisional family?

207 Please define full representation. Is it three sequences/species? Is it one? IS it all of them?!

222 A short word on discrepancy between OTU number assigned (678) and the LFF database with only 600 spp. Does more than one OTU match some species?

But before you get to that, here it becomes rather important to understand how your 1007 OTUs were defined. I assume this is a standard setting in FROGS, but you should let your readers know briefly how the clustering works in the pipeline.

248 Not really sure what this rate describes. ‘within 23.6 species’ is not explained well enough to follow here. Could you just do a worked example here for us? For me, 67.4% identification success makes sense, but not the part about ‘within species’.

244-49 ‘UNITE identification success based on families’ – I think your description could be clearer. What about ‘identification success across tested families’ would be clearer. I had to go back to the methods again here.

303-304 This is standard practice in dietary studies and is known to improve species IDs.

363 missing ‘the’ before nuclear

Supplementary File note: It is interesting the see the span of similarity scores by BLAST - some of them rather low, but a range. I think it would be really valuable to add in a short dissection and discussion of these results, especially if some of the ones with lover than 90% BLAST matches come from vouchers and you're able to see what the issue is, e.g. nothing in that genus has been sequenced, the ITS copy that was sequenced is divergent, whatever. 

Reviewer 2 Report

The authors established a custom database comprised over 4800 sequences and represented over 600 formal and provisional species of lichen-forming fungi and proved that the custom database can significantly improve the species-level assignment of OTUs than the UNITE database.  The work done in the manuscript is significant and worth publishing.

Little problems should be amended:

1. Line 42, after 6.28 million, the reference should include [13-14] , since [14] did not mention the 6.28 million.

2. Line 43, 'funal' should be "fungal".

3. After line 147, should add the PCR condition.

4. Line 314, reference [25, 26, 28, 48-55] inippropriately self-citation, the authors should delete part of them.

5. Line 475, 515, 520, 583, the references format shou be amended

The language is fluent, a few word spelling mistake
